# Serum and Urinary Biomarkers in COVID-19 Patients with or without Baseline Chronic Kidney Disease

**DOI:** 10.3390/jpm13030382

**Published:** 2023-02-22

**Authors:** Rumen Filev, Mila Lyubomirova, Julieta Hristova, Boris Bogov, Krassimir Kalinov, Dobrin Svinarov, Lionel Rostaing

**Affiliations:** 1Department of Nephrology, Internal Disease Clinic, University Hospital “Saint Anna”, 1750 Sofia, Bulgaria; 2Faculty of Medicine, Medical University Sofia, 1504 Sofia, Bulgaria; 3Department of Clinical Laboratory, University Hospital “Alexandrovska”, 1431 Sofia, Bulgaria; 4Head Biometrics Group, Comac-Medical Ltd., 1404 Sofia, Bulgaria; 5Nephrology, Hemodialysis, Apheresis and Kidney Transplantation Department, Grenoble University Hospital, 38043 Grenoble, France; 6Medicine Faculty, University of Grenoble Alpes, 38000 Grenoble, France

**Keywords:** COVID-19, biomarkers, chronic kidney disease, mortality, acute kidney injury, uNGAL, IL-6, uKIM1, CT severity score

## Abstract

In a prospective, observational, non-interventional, single-center study, we assessed various plasma and urinary biomarkers of kidney injury (neutrophil gelatinase-associated Lipocain [NGAL], kidney-injury molecule-1 [KIM-1], and interleukin-18 [IL-18]); inflammation (IL-6, C-reactive protein [CRP]); plus angiotensin converting enzyme 2 (ACE2) in 120 COVID-19 patients (of whom 70 had chronic kidney disease (CKD) at emergency-department (ED) admission). Our aim was to correlate the biomarkers with the outcomes (death, acute kidney injury [AKI]). All patients had received a chest-CT scan at admission to calculate the severity score (0–5). Biomarkers were also assessed in healthy volunteers and non-COVID-19-CKD patients. These biomarkers statistically differed across subgroups, i.e., they were significantly increased in COVID-19 patients, except for urinary (u)KIM1 and uIL-18. Amongst the biomarkers, only IL-6 was independently associated with mortality, along with AKI and not using remdesivir. Regarding the prediction of AKI, only IL-6 and uKIM1 were significantly elevated in patients presenting with AKI. However, AKI could not be predicted. Having high baseline IL-6 levels was associated with subsequent ventilation requirement and death. The mortality rate was almost 90% when the chest CT-scan severity score was 3 or 4 vs. 6.8% when the severity score was 0–2 (*p* < 0.0001).

## 1. Introduction

The SARS-CoV-2 2019 (COVID-19) pandemic has killed millions of people worldwide (https://en.wikipedia.org/wiki/Template:COVID-19_pandemic_data (accessed on 25 August 2022). From a medical point of view, the situation in the Republic of Bulgaria was not that different from the rest of the world; however, the statistics reveal disturbing tendencies, i.e., the percentage of vaccinated people in Bulgaria has been one of the lowest in Europe and the mortality rate has been one of the biggest in the European Union [1].

SARS-CoV-2 infection disproportionately affects patients with hypertension, diabetes mellitus, chronic kidney disease (CKD), and/or cardiovascular disease, as well as older people [2,3,4]. Acute kidney injury (AKI) may be caused by many factors, including heart failure, sepsis, hemorrhage, nephro-toxic drugs, as well as by COVID-19. A recent review paper addressed COVID-19-associated AKI [5]. It is now widely believed that AKI is a frequent and significant complication of COVID-19; however, there is significant variability in its reported incidence and outcomes.

For example, in 2020, Hirsh et al. reported a high prevalence of AKI among hospitalized patients with COVID-19 (36.6%) [6]. In non-COVID-19 patients, it is estimated that 20% of hospitalized patients deteriorate to AKI and 10% of AKI patients require renal-replacement therapy (RRT). The mortality rate of patients requiring RRT is as high as 50%. Patients recovering from AKI have a higher risk of chronic kidney disease and even end-stage renal disease [7,8,9,10]. In those infected with COVID-19, major observational studies and meta-analyses have reported AKI incidence rates of 28–34% in inpatients and 46–77% in those in the intensive care unit (ICU). Nevertheless, the prevalence of more severe cases of AKI that require RRT in the ICU appears to have reduced over time: the data from England and Wales show that RRT has declined from 26% at the start of the pandemic to 14% in 2022 [5].

The risk factors for AKI in COVID-19 are similar to those outside of COVID-19; the presence of underlying CKD is a particularly significant risk factor. Additionally, the existence of new-onset AKI in the ICU independently predicted higher mortality in more severely ill patients, but not in those who experienced the most severe sequential organ failure assessment (SOFA) scores [11]. Thus, these findings show that the presence of hospital-acquired AKI serves as an early indicator of developing systemic multiorgan failure and of subsequent death. Recently, Sabaghian et al. published a systematic review (44 studies) on COVID-19 and AKI. They identified that the most prevalent comorbidities in patients with COVID-19 and suffering from AKI were diabetes, hypertension, and hyperlipidemia. Of the 44 included studies, twelve reported a history of chronic kidney disease (CKD). The most frequent underlying pathological conditions were focal segmental glomerulosclerosis and acute tubular necrosis. The average length of hospital stay was 19 days and the median duration of need for mechanical ventilation was three days. They concluded that AKI often complexed the COVID-19 hospitalization course and was associated with an increase in disease severity, prolonged duration of hospitalization, and poor outcome [12].

Diagnosing AKI early and taking proper preventive and therapeutic measures can effectively improve recovery after AKI. At present, the standard for the diagnosis of AKI is based on the serum creatinine level and urine excretion volume, in accordance with the recommendations issued by the Kidney Disease Improving Global Outcomes (KDIGO) in 2012 [13]. However, serum creatinine and urine output are unspecific and may be delayed, thus obscuring the early diagnosis of AKI. More effective and potential early biomarkers have been discovered with the in-depth studies of AKI pathology, such as neutrophil gelatinase-associated lipocalin (NGAL) in the blood (sNGAL) or urine (uNGAL), urinary kidney-injury molecule-1 (KIM-1), and interleukin-18 (IL-18), serum cystatin C (sCysC), plasma tissue inhibitor of metallo-proteinase 2 (TIMP-2), monocyte chemo-attractant protein (MCP-1), etc. Recently, Menez et al. evaluated 19 urinary biomarkers of injury, inflammation, and repair in 153 hospitalized COVID-19 patients. They found that a twofold higher level of NGAL, MCP-1, and KIM-1 was associated with the highest risk of sustaining a primary composite outcome, i.e., KDIGO stage 3 AKI, requirement for dialysis, or death within 60 days of hospital admission [14]. In a series of 52 COVID-19 patients, of whom 22 developed AKI, and of which 8 required RRT, Pode Shakked et al. found that the serum creatinine and sCysC measured at presentation into an emergency department (ED) were both highly accurate predictors of AKI and the need for RRT, whereas sNGAL demonstrated an adequate diagnostic parameter [15]. In a prospective observational study of 57 COVID-19 patients admitted into intensive care, Luther et al. assessed urinary albumin, NGAL, KIM-1, and plasma TIMP-2 at admission: the majority (89%) of patients developed AKI. They found that biomarkers in the urine were increased in the majority of patients, but they did not predict the KDIGO stage reliably [16].

In this prospective, observational, non-interventional, single-center study, we assessed various plasma and urinary biomarkers of kidney injury (NGAL, KIM-1, IL-18); inflammation (IL-6, C-reactive protein [CRP]); and angiotensin-converting enzyme 2 (ACE2) in COVID-19 patients at ED admission, with the aim of correlating them with the outcomes (mortality, AKI). We also assessed these biomarkers in healthy volunteers and non-COVID-19 CKD patients.

## 2. Patients and Methods

This is a single-center study conducted at the Alexandrovska Hospital in Sofia (Bulgaria) between February 1 and March 31 2021. In summary, we enrolled consecutive patients who had positive COVID-19 infection and were admitted to the hospital after a positive PCR test for SARS-CoV-2. All cases were validated using a reverse transcription-polymerase chain reaction of combined throat/nose samples. In this study, we only included patients >18 years of age. Patients with coexisting urinary tract infection were excluded [17].

Of the 120 patients enrolled into our study, 70 had a history of chronic kidney disease (i.e., impaired kidney function with an estimated glomerular-filtration [eGFR] rate of <60 mL/min, though >20 mL/min, i.e.) and none had end-stage renal disease. The eGFR was estimated using the chronic kidney-disease (CKD)-EPI 2021 formula. The other 50 patients had no history of kidney disease and had normal levels of serum creatinine (i.e., females 44–80 µmol/L; males 62–106 µmol/L; eGFR > 60 mL/min/1.73 m^2^); that group included five patients that had undergone renal transplantation (eGFR of >60 mL/min).

Data on sex, age, comorbidities, and laboratory results of blood drawn were collected on admission to the Emergency Department. The follow-up was performed at 7–10 days after admission. Medical comorbidities, i.e., hypertension, obesity (i.e., body mass index >30 kg/m^2^), diabetes mellitus, vascular disease, and CKD, which should have already been diagnosed along with the medical history, were recorded. When AKI was present, it was categorized according to the KDIGO criteria.

For all patients with a positive COVID-19 test, a chest CT-scan was performed to assess the disease severity and progression. We used simple chest computed-tomography (CT) scores to evaluate the severity of the lung involvement in COVID-19 patients. Each patient was classified between 0–5: score 0 (0% or none), score 1 (1–5% or minimal), score 2 (6–25% or mild), score 3 (26–49% or moderate), score 4 (50–74% or severe), and score 5 (≥75% or extensive) [18].

Upon admission into the ED, the patients were informed of the protocol by one of us (R.F.) and, after they had given their informed consent and signed to give a blood sample, the blood and urinary samples were sent to the laboratory to be stored, frozen, until biomarker analyses.

In total, five different biomarkers were assessed: urinary NGAL, KIM-1, IL-18, serum IL-6, and angiotensin-converting enzyme 2 (ACE2). The determination of NGAL was performed with ELISA kits obtained from Thermo Fisher Scientific, while the assays for IL-6, IL-18, and ACE-2 were performed with ELISA kits obtained from BioVendor R and D. KIM-1 was assayed using ELISA kits from MyBioSource, Inc. All of the analyses were accomplished according to the producers’ instructions. For every biomarker, all the patients’ samples were thawed and assessed the same day by the same technician.

The biomarkers were also assessed in two control groups: 20 healthy volunteers taking no medication and 20 CKD patients. The latter patients needed to have a well-characterized CKD and were followed-up in our outpatient nephrology clinic. In addition, they needed to be in a stable condition. After having given their informed consent to participate in the study, both the healthy volunteers and CKD controls had a PCR-test for SARS-CoV-2 that had to be negative.

For testing ACE-2 (from serum), we used the following kit: Catalogue number RAG006R, produced by: BioVendor–Laboratorní medicína a.s. Karásek 1767/1, 621 00 Brno, Czech Republic. The normal ranges are 0.0625 ng/mL–4 ng/mL. For the IL-6 (from serum) assessment, a kit from Catalogue number RD194015200R (produced by BioVendor–Laboratorní medicína a.s. Karásek 1767/1, 621 00 Brno, Czech Republic) was used. Probes from all the healthy control patients were assessed: the lowest standard result was 1.25 pg/mL, and three samples were measured between 1.25 and 5 pg/mL. Regarding NGAL (from urine), we used the following kit: Catalogue number BMS2202, produced by Thermo Fisher Scientific, Campus Vienna Biocenter 2, 1030 Vienna, Austria. The measured values of our healthy controls were 21–48 ng/mL. With respect to KIM-1 (from urine), we used the kit from Catalogue number MBS020924, produced by: MyBioSource, Inc., San Diego, CA 92195-3308, USA. The measured values from our healthy controls were 1.5–2.1 ng/mL. Finally, for interleukin-18 (from urine), we used the kit from Catalogue number RAF143R, produced by: BioVendor–Laboratorní medicína a.s. Karásek 1767/1, 621 00 Brno, Czech Republic. The measured values from our healthy controls were 27.1–100.5 pg/mL plus one outlier (with a result of 189.7 pg/mL).

The study was conducted according to the guidelines of the Declaration of Helsinki and was approved by the ethical committee KENIMUS at the Medical University of Sofia, Bulgaria, with Protocol №12/31.05.2022. All of the data are available upon request from the corresponding author.

### 2.1. Statistical Analysis

Categorical parameters were described by absolute and relative (percentage) frequencies. Continuous parameters were described by arithmetic means and standard deviations (SD), and median, minimum, and maximum values. The distribution of the continuous parameters was checked for normality using the Shapiro–Wilk test.

### 2.2. Methods for Testing the Post hoc Hypotheses

To compare the continuous parameters in two related (paired) groups, the Wilcoxon two-sample test was applied, and the approximation of the Student’s t-statistic (t-approximation) with a continuity correction of 0.5 was used to determine statistical significance. In addition, the sign test was used where appropriate.

Comparisons between the independent (unrelated) groups (CKD vs. no CKD) were made after adjustment for baseline differences.

Due to the heterogeneity and non-normality of the data distribution, non-parametric ANOVA (Kruskal-Wallis) was used to compare more than two groups.

The linear association between the continuous normally distributed variables was estimated by Pearson’s correlation coefficient. For the continuous non-normal parameters, Spearman’s coefficient was used. Bi-serial coefficient was used to explore the relation between the non-metric variables.

### 2.3. Method Used for Data Modelling

Binary logistic regression was used to model the relationship between the output (COVID-19-related death) as a dependent variable and the main parameters. A model is presented for death. In addition, the odds ratios and forest plots are shown. For decision making, a significance level of 5% was used.

The SAS^®^ package version 9.4 (SAS Institute Inc., SAS 9.4 Help and Documentation, Cary, NC, USA: SAS Institute Inc., 2015–2022) was used for the calculations and the graphical presentations.

## 3. Results

From the total of 160 persons, 75% were confirmed with COVID-19 (120 patients in total), of which 70 (58.3%) had a history of CKD. The other 40 persons (25%) that had no history of COVID-19 were used as controls and were divided into two separate groups: 20 patients (50%) had a history of CKD, whereas the other 20 (50%) were totally healthy (i.e., no history of CKD or any other type of co-morbidity).

The median age of the CKD patients that had a positive test for COVID-19 was 56.8 years, whereas for the non-CKD COVID-19 patients, it was 65.9 years. The gender ratio was 50% in both groups (see Table 1). For the CKD patients without COVID-19, the median age was 66.1 years; the gender ratio was 11 females (55%) to 10 males (45%). For the healthy control group, the median age was 36.8 years, and the gender ratio was equal: 10 male and 10 female patients. Overall, in the COVID-19 patients, the serum-creatinine level on admission was 119.0 μmol/L (57.0–930.0 μmol/L) for the CKD group and 79.0 μmol/L (50.0–295.0 μmol/L) for the non-CKD group. In addition, it was elevated over the baseline in 58.3% of cases.

For the non-COVID-19 CKD patients, the median serum creatinine was 109.1 μmol/L (62.0–188.0 μmol/L); it was elevated in 60% of cases. No patients in the healthy control group had creatinine levels over the baseline, i.e., normal ranges for males are 62–106 μmol/L and for females 44–80 μmol/L. The mean value of eGFR on admission was 80.4 mL/min/1.73 m^2^ for the non-CKD COVID-19 patients, whilst it was 47.9 mL/min/1.73 m^2^ for the CKD COVID-19 patients (*p*-values < 0.0001). For the CKD patients without COVID-19, the eGFR was 62.3 mL/min/1.73 m^2^, whilst the eGFR was at 111.1 mL/min/1.73 m^2^ for the healthy control group.

There were significant differences (<0.0001) between the CRP values across the four groups. All of the COVID-19 patients had elevated CRP levels: 75.7 mg/L (SD = 73.7) in the CKD group and 53.9 mg/L (SD = 59.5) in the non-CKD group. For the COVID-19-negative patients, CRP was at 7.0 mg/L (SD = 6.7) in the CKD group and 1.7 mg/L (SD =1.6) in the healthy controls. The upper limit for CRP was ~5 mg/L.

The five biomarkers statistically differed across the four subgroups (see Table 2); thus, we made comparisons between each subgroup using paired Kruskal-Wallis analysis.

We found the following statistically significant changes:-**CKD and COVID-19 vs. healthy controls**—significant changes for IL-6, ACE2 enzyme, NGAL, and IL-18-**Non-CKD and COVID-19 vs. healthy controls**—significant changes for IL-6, NGAL, and IL-18-**CKD only vs. healthy controls**—significant changes for IL-6, ACE2 enzyme, and IL-18-**CKD and COVID-19 vs. CKD only**—significant changes for ACE2 enzyme and NGAL

These results can be seen in Table 3 and Figure 1.

Acute kidney injury was a frequent event, i.e., it occurred in 38 patients (27.3%), but was registered only in the COVID-19 positive patients, of whom 31 were in the CKD group (44.3%), and 7 (14%) were in the non-CKD group (*p*-value = 0.0006). All of the patients that had AKI could be divided into three stages, using the KDIGO criteria: from all the patients with AKI, 30 patient (i.e., 78.9%) had AKI Stage 3; 4 patients had AKI Stage 2 (10.5%); while the remainder (3 patients or 7.8%) had AKI stage 1. From our analysis, the AKI was assessed as a risk factor in connection to the mortality rate: overall, the hospital mortality was 14.4% (23 patients) of the 160 patients in our group, of which 19 patients, or 82.6% (*p* = 0.009), had AKI (any stage by KDIGO).

Analysis was performed to assess the correlations between the biomarkers and the incidence of AKI in each of the COVID-19 groups. Amongst the biomarkers, we found that the levels of IL-6 (*p* = 0.006) and uKIM-1 (*p* = 0.03) were significantly higher in those patients that presented with AKI, whilst the results for the uNGAL were significantly lower (*p*-value = 0.0002). Conversely, there was no significance for the other two biomarkers: ACE2 enzyme (*p* = 0.62) and uIL-18 (*p* = 0.38) (Figure 2). However, none of the biomarkers were able to predict AKI.

Further correlation analyses were performed with all five biomarkers in relation to the inflammation biomarkers and the degree of lung involvement, thus determining whether any of these criteria were associated with a fatal outcome (Table 4).

The most significant results were found for IL-6, which correlated with the number of leukocytes and neutrophils, and with the degree of lung involvement (as assessed by the Severity Score, ventilation, and mortality). The only non-significant result for IL-6 was with CRP. The biomarkers KIM-1 and CRP also showed a significant correlation. IL-18 showed an association with the neutrophil count and the extent of lung involvement.

We observed that having high baseline IL-6 levels was associated with subsequent ventilation requirement. Indeed, the patients that required subsequent ventilation had IL-6 levels ranging between 25 and 120 pg/mL (median 45.2 pg/mL); there were two outliers under 25 pg/mL. In the COVID-19 patients, 27 patients had IL-6 results of >25 pg/mL on admission and, later, 19 required mechanical ventilation (70.4%).

All of the patients that did not survive the SARS-CoV-2 infection were in the ICU and died from advanced pneumonia; none of these patients had a history of any kind of cardiovascular incident while in the ICU.

All of the COVID-19 patients had received a chest CT-scan upon AD admission. Every CT-scan was graded according to the severity score (0–5). For our patients, the results ranged between 0–4. We grouped the patients into two categories: 0–2 score (n = 102) and 3–4 (n = 18).

We included the following factors in the logistic regression model as important: Gender, Hypertension, AKI, Treatment with Remdesivir, Febrile, CKD, age, IL-6, CRP, uNGAL, uKIM-1, Diabetes, and Proteinuria.

The logistic regression for the mortality risk factors showed that the statistically significant factors were AKI, IL-6 levels, and not being treated with remdesivir (see Figure 3). AKI was a negative prognostic factor for COVID-19 infection, i.e., it increased the mortality [OR = 5.246 (95% CI: 1.279–21.508); *p* = 0.02], as was not being treated with remdesevir [OR = 4.319 (95% CI: 1.104–16.904); *p* = 0.04] and having increased levels of IL-6 [OR = 1.023 (95% CI: 1.002–1.045); *p* = 0.03]. The other biomarkers (KIM-1, and NGAL) included in the model did not predict death.

## 4. Discussion

In this observational, prospective study that included COVID-19 patients with or without CKD, we evaluated the added value of blood and urinary biomarkers assessed at hospital admission with the aim to determine whether they could predict subsequent death and AKI. These biomarkers included urinary NGAL, KIM-1, IL-18, and serum IL-6, CRP, and ACE2. Only the IL-6 level was independently associated with death. A recent systematic review and meta-analysis has shown that IL-6 significantly increases the risk of COVID-19 severity (adjusted OR = 1.0284; 95% CI 1.0130–1.0441; *p* = 0.0003) and mortality (aOR = 1.0076; 95% CI 1.0004–1.0148; *p* = 0.04; adjusted hazard ratio (aHR) = 1.0036; 95% CI 1.0010–1.0061; *p* = 0.006) [19]. Another systematic review of 147 studies has shown that deceased COVID-19 patients had 42.1 times higher mean concentrations of IL-6 than patients that survived. The IL-6 level was significantly increased in those that died (MD: 42.11; *p* < 0.001; 95% CI: 36.86, 47.36) [20].

We also found that not giving remdesivir therapy upon admission into the ED independently increased the risk of death by more than four-fold (OR = 4.319; 95% CI. 1.104–16.904; *p* = 0.04). Indeed, a recent randomized controlled trial has demonstrated that among non-hospitalized COVID-19 patients that had a high risk for its progression, a three day course of remdesivir had an acceptable safety profile and resulted in an 87% lower risk of hospitalization or death than a placebo [21]. In addition, a recent systematic review and meta-analysis has shown that there is a high probability that remdesivir reduces mortality for nonventilated patients with COVID-19 and that require supplemental oxygen therapy [22]. However, the World Health Organization (WHO) Solidarity Trial Consortium has demonstrated that remdesivir has no significant effect on patients with COVID-19 who were already being ventilated [23].

We found that AKI independently increased, by more than 5-fold (OR 5.246 (95%CI 1.279–21.508); *p* = 0.02), the risk of COVID-19-associated death. A recent study from the United States (US) has shown that out of a total cohort of 306,061 COVID-19 patients, 126,478 (41.0 %) had AKI. The AKI patients had higher mortality compared to those without AKI; the incidence of AKI was highest at the beginning of the pandemic (49.3%), but later reduced (40.6%). The severity of AKI was also associated with mortality [24]. Similarly, the incidence of more severe AKI that requires renal-replacement therapy in an ICU appears to have declined over time: the data from England and Wales show that RRT declined from 26% at the start of the pandemic to 14% in 2022 [5].

In our study, we also looked for early biomarkers that might predict COVID-19-associated AKI. Urinary interleukin-18 (uIL-18), as well as ACE2, were not associated with AKI. A recent systematic review evaluated the value of uIL-18 to predict AKI. The estimated sensitivity and specificity of uIL-18 for the diagnosis of AKI were 0.64 (95% confidence interval (CI): 0.54–0.73) and 0.77 (95%CI: 0.71–0.83), respectively. Subgroup analysis showed that uIL-18 in pediatric patients was more effective at predicting AKI than in adults [25]. Saygili et al. assessed uIL-18 in 71 COVID-19 children, of whom 12 had AKI. Compared to healthy control children, uIL-18 was significantly increased; however, only 4 of 12 AKI children had increased levels of uIL-18 [26]. The actions of angiotensin-converting enzyme 2 (ACE2) oppose those of the renin-angiotensin-aldosterone system. The SARS-CoV-2 cellular entry receptor is ACE2. ACE2 may be a cytoprotectant in some tissues. It has been shown that renal ACE2 expression is decreased in ischemic AKI [27]. We found that the ACE2 blood levels were significantly higher in the COVID-19 patients compared to the non-COVID-19 patients and healthy volunteers. In addition, amongst the CKD patients, the ACE2 levels were significantly higher in the COVID-19 positive group compared to the COVID-19-negative patients.

We found that the IL-6 levels were significantly higher in the COVID-19 patients and in the non-COVID-19 CKD patients compared to the healthy volunteers. In the COVID-19 patients, those that had AKI had significantly higher IL-6 levels compared to those that had no AKI (20.4 vs. 5.53 pg/mL). Wang et al. have shown that the L-6 level had significant positive correlations with serum creatinine and blood urea nitrogen [28]. In the patients with COVID-19, the serum levels of IL-6 were elevated in those with AKI [29]. In addition, the serum levels of IL-6 can also predict the clinical outcomes of AKI as it is significantly reduced in those where AKI is eliminated after effective treatment [30]. Finally, levels of IL-6 of >35 pg/mL may indicate a risk of respiratory failure [31] in the context of a COVID-19 infection. Herein, we confirm that result, i.e., out of 27 COVID-19 patients that had IL-6 results of >25 pg/mL at admission, later, 19 (70.4%) of these required mechanical ventilation.

KIM1 expression is dramatically up-regulated in kidney post-ischaemia/reperfusion injury in rats, as well as in rodent models of drug-induced AKI. Its expression is mainly upregulated in proximal tubule cells, both in rodents and in man. KIM-1 plays an important role in kidney recovery and tubular regeneration [32]. In a prospective study, 19 urinary biomarkers were assessed to predict AKI in COVID-19 patients; among them, twofold higher levels of NGAL (HR, 1.34 [95% CI, 1.14–1.57]), monocyte chemoattractant protein (MCP-1) (HR, 1.42 [95% CI, 1.09–1.84]), and KIM-1 (HR, 2.03 [95% CI, 1.38–2.99]) were associated with the highest risk of sustaining a primary composite outcome (KDIGO stage 3 AKI, requirement for dialysis, or death within 60 days of hospital admission) (14). However, individual biomarkers provided moderate discrimination, and biomarker combinations improved the discrimination from the primary outcome. We observed that uKIM1 in the COVID-19 patients was significantly higher in those that had AKI versus the others; however, the absolute difference was very small (1.95 vs. 1.85 ng/mL *p* = 0.03).

Neutrophil gelatinase-associated lipocalin (NGAL) concentrations in urine or plasma may identify patients with a high risk for AKI in clinical research and practice [33]. A recent systematic review has shown that NGAL appeared to have a predictive value irrespective of age, from newborn to 78 years. NGAL levels can accurately predict the outcome and severity of AKI occurring in several disease processes, including contrast-induced AKI during cardiac surgery, kidney-transplant rejection, chronic heart failure, and systemic inflammation in critically ill patients, even though the significance of NGAL is highly variable across studies [34]. In the setting of COVID-19 patients, a prospective cohort observational study consisting of 444 consecutive patients evaluated uNGA at hospital admission. They found that the levels were associated with AKI diagnosis (267 ± 301 vs. 96 ± 139 ng/mL, *p* < 0.0001) and staging; uNGAL levels >150 ng/mL had 80% specificity and 75% sensitivity to diagnose AKI stages 2 to 3. At admission, the uNGAL level was quantitatively associated with prolonged AKI, dialysis, shock, prolonged hospitalization, and in-hospital death, even when the admission serum-creatinine level was not elevated [35].

Conversely, low uNGAL levels at admission ruled out stages 2–3 AKI (negative predictive value: 0.95, 95% CI: 0.92–0.97) and the need for dialysis (negative predictive value: 0.98, 95% CI: 0.96–0.99). In contrast, in a prospective cohort of 153 COVID-19 patients, Menez et al. evaluated the utility of urinary biomarkers such as NGAL, KIM1, MCP1 to predict AKI. They observed that individual biomarkers provided moderate discrimination and biomarker combinations improved the discrimination for the primary outcome [14]. In our study, we found that the uNGAL in healthy controls and in non-COVID-19 patients was very similar; however, it was almost five times higher in those with COVID-19. However, surprisingly, the uNGAL was significantly lower in those that will develop AKI as compared to those that will not (115.48 vs. 218.47 ng/mL; *p* = 0.0002). We have no explanation for this unexpected result.

We looked at whether the biomarkers we evaluated were associated with the chest-CT severity score (CTSS). Valk et al. found that, despite a poor prognostic capacity, CTSS was associated with ICU mortality [36]. Almasi Nokiani et al. found that CTSS is an excellent tool in triage and prognostication in patients with COVID-19 aged ≥65 years, but is of limited value for younger patients [37]. Some studies have found a positive correlation between the extent of CT lung involvement and short-term mortality [38,39]. In addition, there was a significant association between the CTSS and hospital admission, ICU admission, and 30 day mortality [40]. In addition, patients with a score of three had a higher risk for complications and a fatal outcome [41]. In our study, we found that the patients with COVID-19 at admission with a CTSS score of —between three and four had a mortality rate of 88.9%, compared to only 6.8% in those where the CTSS was —between zero and two (*p*-value < 0.0001). In addition, we found that some biomarkers were significantly associated with the CTSS score, i.e., uIL-18, uNGAL, and serum IL-6. In the literature, we were not able to find other such correlations. Herold et al. reported that the maximal level of IL-6, followed by the CRP level, was highly predictive for the need of mechanical ventilation, but these patients did not have a chest CT-scan at admission [31]. Therefore, taking into account all of our results, we found that patients that had higher IL-6 levels on admission and had a chest CT severity score of >3 had a much greater risk of needing mechanical ventilation. Sadly, all of the ventilated patients from our group have passed away.

Our study has some limitations: we only had 120 COVID-19 patients; however, they were very well phenotyped. In addition, the biomarkers were only assessed at admission in the ED, but not subsequently; this would have been of interested in order to study their kinetics. This might explain, in particular, why they were of very limited value, except with the IL-6 levels.

## 5. Conclusions

In this prospective case-series of 120 COVID-19 patients, the assessment at ED admission of the urinary biomarkers for kidney injury (NGAL, KIM-1, IL-18) was of limited value to predict AKI. Conversely, serum IL-6 was an independent predictor of death. Finally, IL-6 levels were associated with needing ventilation.

## Figures and Tables

**Figure 1 jpm-13-00382-f001:**
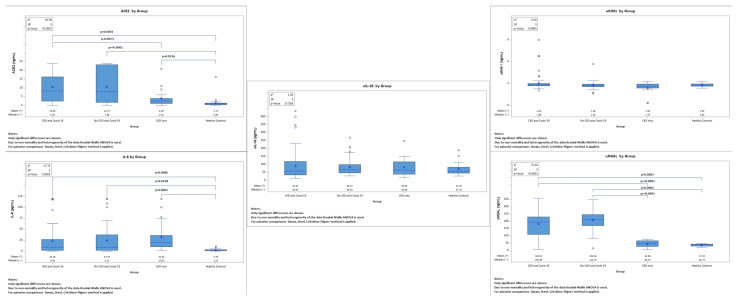
Comparisons between biomarkers across the four different groups.

**Figure 2 jpm-13-00382-f002:**
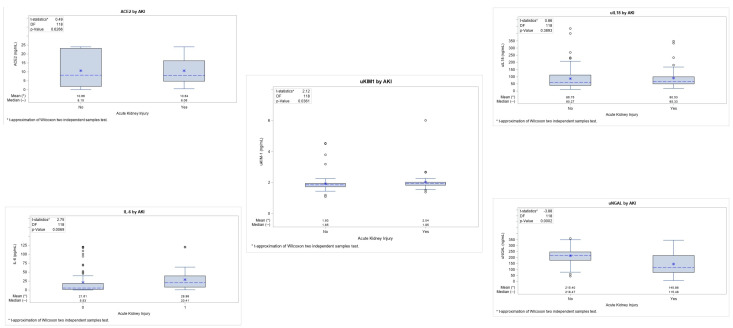
Comparison of biomarkers in COVID-19 patients with or without AKI.

**Figure 3 jpm-13-00382-f003:**
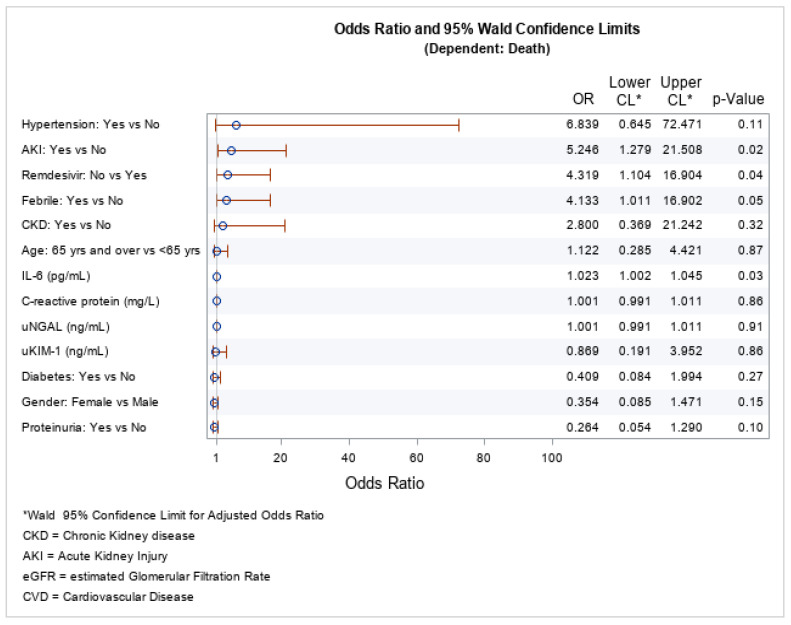
Odds ratio for risk factors for COVID-19-related death.

**Table 1 jpm-13-00382-t001:** Population description.

Group	Number of Patients	Gender	Age (yrs) Mean (SD)	Race (*n*%)	Creatinine (μmol/L) Mean SD	eGFR mL/min/1.73 m^2^ Mean (SD)	CRP (mg/L) Mean (SD)
CKD^+^ COVID-19^+^	70	Female: 21 (42.0%)	56.8 (16.0)	Caucasians 100%	171.4 (156.4)	47.9 (23.0)	75.7 (73.7)
		Male: 29 (58.0%)					
Non-CKD COVID-19^+^	50	Female: 39 (55.7%)	65.9 (12.6)	Caucasians 100%	120.8 (176.9)	80.4 (28.9)	53.9 (59.5)
		Male: 31 (44.3%)					
CKD without COVID-19	20	Female: 11 (55%)	66.1 (11.8)	Caucasians 100%	109.1 (31.8)	62.3 (22.6)	7.0 (6.7)
		Male: 9 (45%)					
Healthy controls	20	Female:10 (50%)	36.8 (7.8)	Caucasians 100%	68.8 (15.6)	111.1 (13.0)	1.7 (1.6)
		Male:10 (50%)					
*p*-values		0.49	<0.0001	NA	0.02	<0.0001	<0.0001

Abbreviation: CKD: chronic kidney disease; CRP: C-reactive protein; eGFR: estimated glomerular-filtration rate; SD: standard deviation. + shows that the patients are possitive for COVID-19 infection.

**Table 2 jpm-13-00382-t002:** Biomarkers by groups by median (ranges).

	Biomarker	IL-6 *Median* *(Ranges)*	ACE-2 *Median* *(Ranges)*	uKIM-1 *Median* *(Ranges)*	uNGAL *Median* *(Ranges)*	uIL-18 *Median* *(Ranges)*
Group						
CKD and COVID-19		9.45 (0.22–120.0)	8.30 (0.09–24.0)	1.89 (1.38–6.00)	195.96 (7.80–359.79)	58.97 (12.50–436.35)
Non-CKD and COVID-19		8.15 (0.28–120.0)	7.98 (0.14–24.00)	1.84 (1.10–3.79)	214.75 (15.21–349.60)	66.35 (28.17–269.19)
CKD without COVID-19		20.62 (1.75–120.0)	2.54 (0.06–21.02)	1.75 (0.25–2.20)	49.75 (5.66–76.69)	64.96 (16.72–248.81)
Healthy Controls		2.52 (0.2–10.6)	0.95 (0.04–16.35)	1.87 (0.036–16.35)	40.73 (21.17–48.45)	57.78 (27.66–189.76)
Chi-square * DF *p*-values		17.74 3 0.0005	35.58 3 <0.0001	6.53 3 0.0884	75.16 3 <0.0001	15.73 3 0.0013

(* Kruskal–Wallis test). *Abbreviations: CKD chronic kidney disease; DF, degree of freedom; IL-, interleukin; u, urinary; ACE2, angiotensin-converting enzyme 2; KIM1, kidney-injury molecule-1; NGAL, neutrophil gelatinase-associated lipocalin.*

**Table 3 jpm-13-00382-t003:** Comparisons between biomarkers (Kruskal–Wallis ANOVA).

Biomarker	Pairwise Comparisons	Statistic *	*p*-Value
**IL-6 (pg/mL)**	CKD and COVID-19 vs. Non-CKD and COVID-19	0.1958	0.99
	CKD and COVID-19 vs. CKD only	2.8622	0.17
	CKD and COVID-19 vs. Healthy Controls	4.4681	0.008
	Non-CKD and COVID-19 vs. CKD only	2.7306	0.21
	Non-CKD and COVID-19 vs. Healthy Controls	4.0085	0.02
	CKD only vs. Healthy Controls	6.0071	0.0001
**ACE2 (ng/mL)**	CKD and COVID-19 vs. Non-CKD and COVID-19	0.4837	0.98
	CKD and COVID-19 vs. CKD only	4.6668	0.005
	CKD and COVID-19 vs. Healthy Controls	7.6153	<0.0001
	Non-CKD and COVID-19 vs. CKD only	3.3897	0.07
	Non-CKD and COVID-19 vs. Healthy Controls	6.2270	<0.0001
	CKD only vs. Healthy Controls	4.2662	0.01
**uKIM-1 (ng/mL)**	CKD and COVID-19 vs. Non-CKD and COVID-19	2.2850	0.36
	CKD and COVID-19 vs. CKD only	3.1093	0.12
	CKD and COVID-19 vs. Healthy Controls	1.5512	0.69
	Non-CKD and COVID-19 vs. CKD only	1.8297	0.56
	Non-CKD and COVID-19 vs. Healthy Controls	0.2023	0.99
	CKD only vs. Healthy Controls	1.8957	0.53
**uNGAL (ng/mL)**	CKD and COVID-19 vs. Non-CKD and COVID-19	2.2998	0.36
	CKD and COVID-19 vs. CKD only	8.1940	<0.0001
	CKD and COVID-19 vs. Healthy Controls	8.3175	<0.0001
	Non-CKD and COVID-19 vs. CKD only	8.8986	<0.0001
	Non-CKD and COVID-19 vs. Healthy Controls	8.8251	<0.0001
	CKD only vs. Healthy Controls	1.6832	0.63
**uIL-18 (pg/mL)**	CKD and COVID-19 vs. Non-CKD and COVID-19	0.1858	0.89
	CKD and COVID-19 vs. CKD only	2.9322	0.18
	CKD and COVID-19 vs. Healthy Controls	4.4732	0.009
	Non-CKD and COVID-19 vs. CKD only	2.9208	0.23
	Non-CKD and COVID-19 vs. Healthy Controls	4.3205	0.03
	CKD only vs. Healthy Controls	6.3278	0.0003

*Abbreviations: ACE2: Angiotensin-converting enzyme 2; CKD: chronic kidney disease; IL-6: Interleukin—6; uIL-18: urinary interleukin—18; uKIM-1: urinary kidney injury molecule; uNGAL: urinary neutrophil gelatinase-associated lipocalin*. *** For pairwise comparisons: the Dwass, Steel, Critchlow-Fligner method was used.**

**Table 4 jpm-13-00382-t004:** Overall correlation for the biomarkers for the COVID-19 positive patients.

	IL-6	ACE2	uKIM1	uNGAL	uIL-18
**Leucocytes *** ***p*-values**	0.18728 0.04	0.09133 0.32	0.02098 0.82	−0.08798 0.33	0.06030 0.51
**Neutrophils *** ***p*-values**	0.35128 <0.0001	0.12588 0.17	0.03270 0.72	0.01535 0.86	0.19583 0.03
**CRP** ***p*-values**	0.01911 0.83	0.14288 0.11	0.23489 0.009	−0.03957 0.66	0.03148 0.73
**CT-Scan **** **(Severity score)** ***p*-values**	0.32857 0.0003	0.06040 0.51	0.11729 0.20	0.19401 0.03	0.29355 0.001
**Ventilation ***** ***p*-values**	0.27175 0.002	−0.08364 0.36	0.11117 0.22	−0.10127 0.27	0.14656 0.11
**Death ***** ***p*-values**	0.27175 0.002	−0.08364 0.36	0.11117 0.22	−0.10127 0.27	0.14656 0.11

*Abbreviations: CT-scan, computed tomography scan; CRP, C-reactive protein; u, urinary; ACE2, angiotensin converting enzyme 2; KIM1, kidney-injury molecule-1; NGAL, neutrophil gelatinase-associated lipocalin*. *** Pearson’s coefficients. ** Spearman’s coefficients. *** Point bi-serial coefficients.**

## Data Availability

The data are available upon request.

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
