# Peer review of "Serum and Urinary Biomarkers in COVID-19 Patients with or without Baseline Chronic Kidney Disease"

_jpm, 2023, doi:10.3390/jpm13030382_

Round 1

Reviewer 1 Report

Study is well defined but rather small number of patients. Study is well presented and could be of additional value to clinicans. 

Author Response

Study is well defined but rather small number of patients. Study is well presented and could be of additional value to clinicians.

No suggestions added by this reviewer!

Reviewer 2 Report

 Filev et al; has performed an observational, prospective clinical study that included COVID19 patients with or without CKD. The authors had evaluated the added value of blood and urinary biomarkers assessed at hospital admission with the aim to determine whether they could predict subsequent death and AKI. These biomarkers included urinary NGAL, KIM-1, IL-18, and serum IL-6, CRP, and ACE2. The study is well defined, performed and analyzed. Here are few comments.

1)     Graphs are not of good quality, they are too small, with small letters. please redo them.

2)     Please do a read through for English language mistakes.

3)     Correct sentence 347 – “deceased COVID-19 patients had 42.1 times higher mean concentration of what?? Than”

Author Response

Filev et al; has performed an observational, prospective clinical study that included COVID19 patients with or without CKD. The authors had evaluated the added value of blood and urinary biomarkers assessed at hospital admission with the aim to determine whether they could predict subsequent death and AKI. These biomarkers included urinary NGAL, KIM-1, IL-18, and serum IL-6, CRP, and ACE2. The study is well defined, performed and analyzed. Here are few comments.

1)     Graphs are not of good quality, they are too small, with small letters. please redo them.

The graphs are with improved quality, easily and clear look at the letters and numbers even at 500% zoom.
Also, separately the images of the graphs are uploaded in the e-mail.

2)     Please do a read through for English language mistakes.

The paper was corrected and read by naturally English-speaking colleague. The same has been done for the newly corrected abstracts in the latest version.

3)     Correct sentence 347 – “deceased COVID-19 patients had 42.1 times higher mean concentration of what?? Than”

Thank you for this comment. Information was added and corrected in the text.

Reviewer 3 Report

Dear authors,

The reviewer's comments are listed below.

1. The Healthy control group is younger than the other groups (CKD and COVID-19, Non-CKD and COVID-19, CKD only), so it may be difficult to compare them with older patients. What are the authors' thoughts on this?

2. Results have been reported that taking RAS inhibitors is not associated with COVID-19 severity or, rather, prevents it. Of the four groups compared in this study (CKD and COVID-19, Non-CKD and COVID-19, CKD only, Healthy controls), are there any patients taking a RAS inhibitor (angiotensin-converting enzyme [ACE] inhibitor or angiotensin receptor blocker [ARB]) Are any patients taking RAS inhibitors (angiotensin-converting enzyme [ACE] inhibitors or angiotensin receptor blockers [ARBs])?

3. It is known that women and men differ in their susceptibility and response to viral infections, resulting in gender differences in incidence and disease severity. Are there any data (4 groups) comparing the evaluation of biomarkers (IL-6, ACE-2, KIM-1, etc.) between men and women in this study?

4. Cytokines induce inflammation as the body's response to infection; in some cases, as seen in COVID-19, cytokines are released in excessive or uncontrolled amounts. This cytokine storm is thought to cause fatal organ failure, including the heart, lung, and kidney systems, and may lead to death. Initial reports from China have shown elevated IL-6, IL-2R, CRP, and ferritin levels in severe cases of SARS-CoV-2 infection (Wenjun Wang Jr, et al. J. Infect. Dis. 2020; 222: 1444-1451, Ruan Q, et al. Intensive Care Med. 2020; 46: 846-848). More detailed studies have confirmed persistently elevated levels of another cytokine, TNF-α, CXCL10, MCP-3, and IL-1RA, in severe cases of COVID-19 (Yang Y et al. J. Allergy Clin. 2020; 146: 119- 127, Pederson SF, et al. J. Clin. Invest. 2020; 130: 2202-2205.). Furthermore, changes in IL-10 correlate with changes in serum creatinine, and some reports indicate that TNF can predict the development of AKI. Why did the authors not include IL-10 or TNF as an item in this evaluation?

Author Response

  1. The Healthy control group is younger than the other groups (CKD and COVID-19, Non-CKD and COVID-19, CKD only), so it may be difficult to compare them with older patients. What are the authors' thoughts on this?

The control group is composed of individuals who meet several conditions:

  1. To be completely healthy, to have no evidence of concomitant diseases.
  2. Have no history of frequent urinary tract infections.
  3. Agree to participate in the study, which must be noted in writing. There was no financial benefit in participation due to the small financial resource, which was mainly directed towards the purchase of biomarkers.

Due to these criteria, this group of patients was recruited as age was not a factor to consider due to the completely voluntary nature of participation.

  1. Results have been reported that taking RAS inhibitors is not associated with COVID-19 severity or, rather, prevents it. Of the four groups compared in this study (CKD and COVID-19, Non-CKD and COVID-19, CKD only, Healthy controls), are there any patients taking a RAS inhibitor (angiotensin-converting enzyme [ACE] inhibitor or angiotensin receptor blocker [ARB]) Are any patients taking RAS inhibitors (angiotensin-converting enzyme [ACE] inhibitors or angiotensin receptor blockers [ARBs])?

Patients taking this group of medications were evaluated in our previous publication, which was entirely related to the use of RAAS blockers - https://doi.org/10.1097/md.0000000000031988
In general 36 patients were taking RAAS blockers (21 women; 15 men). In conclusion, COVID-19 patients with or without CKD who were on RAAS blockade at admission and who were maintained on that therapy during hospitalization had the same outcomes in terms of mortality, ICU admission and AKI rate as those who were not on RAAS blockade. In addition, the RAAS-blocker-treated hypertensive patients had a significantly lower mortality rate.

  1. It is known that women and men differ in their susceptibility and response to viral infections, resulting in gender differences in incidence and disease severity. Are there any data (4 groups) comparing the evaluation of biomarkers (IL-6, ACE-2, KIM-1, etc.) between men and women in this study?

Such a statistical analysis was conducted, but it was without significant results due to which it was not disclosed in the paper.

  1. Cytokines induce inflammation as the body's response to infection; in some cases, as seen in COVID-19, cytokines are released in excessive or uncontrolled amounts. This cytokine storm is thought to cause fatal organ failure, including the heart, lung, and kidney systems, and may lead to death. Initial reports from China have shown elevated IL-6, IL-2R, CRP, and ferritin levels in severe cases of SARS-CoV-2 infection (Wenjun Wang Jr, et al. J. Infect. Dis. 2020; 222: 1444-1451, Ruan Q, et al. Intensive Care Med. 2020; 46: 846-848). More detailed studies have confirmed persistently elevated levels of another cytokine, TNF-α, CXCL10, MCP-3, and IL-1RA, in severe cases of COVID-19 (Yang Y et al. J. Allergy Clin. 2020; 146: 119- 127, Pederson SF, et al. J. Clin. Invest. 2020; 130: 2202-2205.). Furthermore, changes in IL-10 correlate with changes in serum creatinine, and some reports indicate that TNF can predict the development of AKI. Why did the authors not include IL-10 or TNF as an item in this evaluation?

We are thankful for the suggestion from the reviewer. We are aware of the published results on the promising results of the IL-10 study, especially in the context of COVID-19 and diabetes mellitus, as well as for TNF.

The reason we chose to focus on the biomarkers identified was both financial (the cost of shipping the kits to Bulgaria) and the feasibility/availability of conducting the tests in the clinical laboratory of the hospital where the study was conducted.